# Exogenous Salicylic Acid Improves Chilling Tolerance in Maize Seedlings by Improving Plant Growth and Physiological Characteristics

Qian Zhang [†] , Dongmei Li [†], Qi Wang, Xiangyu Song, Yingbo Wang, Xilang Yang, Dongling Qin, Tenglong Xie and Deguang Yang *

College of Agriculture, Northeast Agricultural University, Harbin 150030, China; zhq2016@neau.edu.cn (Q.Z.); ldm2010@neau.edu.cn (D.L.); wq9264741998@163.com (Q.W.); a2737975334@163.com (X.S.); wangyb316@163.com (Y.W.); xilangyang@126.com (X.Y.); a17858198220@163.com (D.Q.); Xie@neau.edu.cn (T.X.)
* Correspondence: ydg2004@neau.edu.cn
† These authors contributed equally to this work.

**Abstract:** Maize (*Zea mays* L.) is a chilling-sensitive plant. Chilling stress in the early seedling stage seriously limits the growth, development, productivity and geographic distribution of maize. Salicylic acid (SA) is a plant growth regulator involved in the defenses against abiotic and biotic stresses as well as in plant development. However, the physiological mechanisms underlying the effects of foliar applied SA on different maize inbred lines under chilling stress are unclear. Two inbred lines, cold-sensitive cv. C546 and cold-tolerant cv. B125, were used to study the effects of SA on the growth and physiology of maize seedlings. The results showed that the application of SA at 50 mg/L on the leaves of maize seedlings under 4 °C decreased the relative electrolyte conductivity (REC) and the malondialdehyde (MDA) and reactive oxygen species (ROS) ($H_2O_2$ and $O_2^-$) content due to increased superoxide dismutase (SOD), peroxidase (POD), catalase (CAT) and ascorbate peroxidase (APX) activity; SA also improved photosynthesis in the seedlings through increased chlorophyll content, enhanced *Pn* and *Gs*, and decreased *Ci*. SA application also increased the proline content and the relative water content (RWC) in the maize seedlings, thereby improving their osmotic adjustment capacity. The increase rate caused by SA of plant height and dry weight in C546 were 10.5% and 5.4% higher than that in B125 under 4 °C. In conclusion, SA promotes maize seedling growth and physiological characteristics, thus enhancing chilling resistance and the effect of SA on the chilling resistance of cold-sensitive cv. was stronger than that on cold-tolerant cv. at the low temperature.

**Keywords:** inbred line; plant growth; photosynthetic characteristics; reactive oxygen species (ROS) content; antioxidant enzyme

## 1. Introduction

Nowadays, global frequent extreme climates and related abiotic stress conditions, such as extreme temperature, drought, waterlogging, salinity, and heavy metals, greatly affect plant growth and development, and influence crop yield and quality. Plants are organisms that cannot escape from adverse environment stress, but only endure. Therefore, plants suffer damage and change during stress, including accumulation of reactive oxygen species (ROS), such as $H_2O_2$, $O_2^-$, and $HO^-$, destruction of structure and function of membrane and chloroplast, reduction in photosynthesis and stomatal conductance [1–5]. However, the damage caused by abiotic stress is alleviated and eliminated by internal defense system in plants, through increasing the accumulation of hormones, such as ABA, methyl jasmonate (MeJAs), salicylic acid (SA) and other osmoprotectants, such as calcium ion ($Ca^{2+}$), proline (Pro) and betaine [1,3,6,7].

Maize (*Zea mays* L.) is one of the most important cereal crops and is used as animal feed, food, and industrial raw material. To increase production to meet the high demand for maize, it has recently been sown in cooler geographical zones. However, maize is native to the tropics, where the plants are very sensitive to chilling [8]. The capacity for maize biomass production decreases in the range 10–15 °C, and these temperatures can even result in irreversible damage; plant death occurs if the plants are exposed below 4 °C for a long period [9]. Low temperatures below 10 °C often occur during seed germination or the seedling stage in early spring. Therefore, cold stress seriously inhibits the growth and development in the seedlings stage of maize, leading to a decline in grain yield and quality [10,11]. Many studies have reported that plants undergo complex physiological and biochemical changes under cold stress [12,13], such as the accumulation of reactive oxygen species (ROS) in subcellular compartments and damage to the cell membrane and photosynthetic systems [14,15].

Salicylic acid (SA) is a phenolic phytohormone that is widely present in plants and is considered to be an endogenous signaling substance [7,16]. It play a vital role as a signaling molecule to regulate the defense system of plants against biotic and abiotic stresses; it also plays roles in plant growth and development, mineral absorption and transportation, flowering, photosynthesis and transpiration [17–19]. Klara [20] reported that abiotic stress promotes the accumulation of free SA and SA glucoside in wheat plants, which shows that SA is involved in regulating the stress response in wheat. Many studies have shown that the exogenous of SA can induce crops to tolerate a variety of abiotic stresses, such as heat, cold, drought, salinity, water, and toxic heavy metal stress [21,22]. The application of SA induced the chilling tolerance of in wheat with the concentration of 100 μM, in *dendrobium officinale* (1.5 mM) and in *brassica napus* (200 μM) [7].

Cold stress remains one of the most destructive abiotic stress factors that significantly affect the maize production and distribution in Northeast China (the Golden Maize Belt). It is important and urgent to find a solution to the cold tolerance of maize. The breeding or cultivation technology are not time-effective or efficient. The quest remains for eco-friendly, cost-effective, and reproducible alternatives, such as exogenous SA, which is inexpensive and easy to apply. Therefore, the application of exogenous SA to improve the cold resistance of crops, could be implemented effectively and more extensively [1]. However, the effects of exogenous SA on cold tolerance in maize are limited reported, especially, the concentration. The researches of the physiological mechanisms underlying the response to exogenous SA in maize inbred lines are also few. Our previous research, where 5 concentrations of SA 0, 25, 50, 100, and 150 mg/L had been set, found that the optimal concentration is 50 mg/L (only published in Chinese) [23]. Therefore, the objectives of this study were to investigate the effects of 50 mg/L SA on the growth and physiological characteristics of two maize inbred lines under cold stress and identify which inbred line was more sensitive to exogenous SA. This study will provide novel prospect into the application and breeding work of SA-induced cold tolerance in maize.

## 2. Materials and Methods

### 2.1. Crop Materials and Experimental Design

We used two maize inbred lines, C546 (cold-sensitive) and B125 (cold-tolerant), for this research. The seed materials were received from Heilongjiang Academy of Agricultural Sciences.

SA (molecular formula: $C_7H_6O_3$, molecular weight: 138) was purchased from Shanghai Sangon Biological Engineering Technology and Services Co., Ltd. (Shanghai, China). Nitroblue tetrazolium, guaiacol, thiobarbituric acid, Coomassie brilliant blue G-250, and other reagents were purchased locally.

In the experiment, we tested the SA concentration of 50 mg/L (the optimal concentration, determined based on the results of a previous experiment), and deionized water as the control treatment (CK), on the same two maize inbred lines. Four treatments were implemented at random, with eight replicates per treatment. The four healthy seeds from

each replication were sown in soil in plastic tubs (21 cm diameter $\times$ 20 cm height). When the maize seedlings had grown three leaves, each plant was sprayed with 10 mL of the SA solution. Deionized water was sprayed as the control treatment. One day after spraying, the maize seedlings were moved into a growth chamber which had a light intensity of 800 mol/m$^2$/s, a day/night cycle of 14 h/10 h, relative humidity of 80% and a temperature of 4 °C, for 2 days. Then, the plants were sampled and examined.

### 2.2. Experimental Measurements

2.2.1. Photosynthetic Characteristics

The photosynthetic rate (*Pn*), stomatal conductance (*Gs*), and intercellular $CO_2$ concentration (*Ci*) of the third complete leaf were determined from 9:30 to 11:00 in the lab, using a portable photosynthesis apparatus (GFS-3000, WALZ, Germany) under a steady light intensity of 800 μmol/m$^2$/s, a relative humidity of 80% and a $CO_2$ concentration of 380 ppm. The area of chamber head was 4 cm$^2$, and temperature in the cuvette was 25 °C.

2.2.2. Relative electrolyte conductivity

The third leaf of the plant was sampled and cleaned. Samples from the leaves were collected with a punch and then placed into triangular bottles with 20 mL re-distilled water for 15 h. The solution was measured using a conductivity meter (FE30, MET). The relative electrolyte conductivity (REC) was calculated according to Zhang [23].

2.2.3. Relative Water Content

Leaf samples were cleaned, and the weight (*fresh wt.*) was recorded. The leaves were placed into triangular bottles with sufficient distilled water for 12 h. Then, the blade surface of each leaf was dried, and the leaf weight was recorded (*turgid wt.*). Each leaf was then baked at 80 °C for 24 h, and the leaf weight was recorded (*dry wt.*). The relative water content (RWC) of the leaves was determined according to Lutts [24].

2.2.4. Proline Content

The proline content was determined using Bates's method [25], and 0.1 g of tissue was homogenized in 10 mL of 3% aqueous sulfosalicylic acid for 10 min followed by filtration. Two milliliters of the filtrate were mixed with 2 mL of acid ninhydrin and 2 mL of glacial acetic acid for 1 h at 90 °C. The developed color was extracted in 4 mL toluene and the intensity of the reaction mixture was determined spectrophotometrically at wavelengths of 520 nm.

2.2.5. Indicators of Oxidative Damage

To measure the MDA content, 2 mL of enzyme solution and 2 mL of 0.67% thiobarbituric acid (TBA) were mixed together, heated in a water bath at 100 °C for 30 min, cooled, and then centrifuged. The absorbance of the supernatant was measured at wavelengths of 450 nm, 532 nm, and 600 nm [26].

The $O_{2-}$ formation rate was determined according to the method of Elstner [27]. First, 0.5 mL of the reaction mixture was combined with 0.5 mL α-naphthylamine solution and 0.5 mL sulfanilic acid solution and shaken until evenly mixed. After being placed at room temperature for 20 min, the optical density of the mixture was determined at wavelengths of 530 nm with a UV-visible spectrophotometer (PERSEE, T6).

The $H_2O_2$ content was determined according to the method of Jana [28]. Then, 0.3 g leaf tissue was homogenized with 3 mL of 50 mM phosphate buffer (pH 6.5). The homogenate was centrifuged at $10,000\times g$ for 25 min. The supernatant was brought to 3 mL with phosphate buffer. The 3 mL extracted solution was mixed with 1 mL of 0.1% titanium sulphate in 20% $H_2SO_4$ (*v/v*), and the mixture was centrifuged at $6000\times g$ for 15 min. To determine the $H_2O_2$ content, the intensity of the yellow color of the supernatant was measured at wavelengths of 410 nm. The $H_2O_2$ content was calculated from a standard curve prepared from hydrogen peroxide of known concentration.

### 2.2.6. Antioxidant Enzyme Activity

Approximately 0.5 g of fresh maize leaves was finely ground in liquid nitrogen.

The SOD activity was determined according to the method of Giannopolitis [29]. The enzyme solution of 20 μL was mixed with 3 mL of the SOD reaction solution. The enzyme solution and control treatments were placed in 74 μmol/m$^2$/s light for 30 min. The absorbance values were recorded at 560 nm.

The POD activity was determined according to the method of Hernandez [30]. The enzyme solution was mixed with 3 mL of the POD reaction solution. The absorbance values were recorded once every 30 s at 470 nm.

The CAT activity was determined according to the method of Sohn [31] with minor modifications as the ability of CAT to consume an amount of $H_2O_2$. Changes in the absorbance of the reaction solution at 240 nm were recorded for 5 min. One unit of CAT activity was defined as an absorbance change of 0.01 units per minute.

The APX activity was determined according to the method of Kang [32]. The homogenate of 3 mL reaction mixture was then centrifuged at $10,000 \times g$ for 20 min. The oxidation of ascorbate was followed by a decrease in the absorbance at 240 nm.

### 2.3. Statistical Analysis

All data were analyzed using the general linear model procedure (GLM) in SPSS 17. Comparisons were made using Duncan's Multiple Range test ($p < 0.05$). The data are shown as the mean ± SE of three replicates.

## 3. Results

### 3.1. Effects of SA on the Growth and Biomass of the Maize Seedlings

The seedling height and root length were significantly influenced by the temperature, inbred line, and SA treatment ($p < 0.001$). The plant height and root length decreased by 8.4% and 13.7%, respectively, in B125 and by 21.2% and 14.1%, respectively, in C546 at 4 °C compared to those at 25 °C (Table 1). C546 was more sensitive to cold than B125, especially its aboveground parts. At the low temperature, SA significantly affected the growth of the inbred lines; however, it had no significant effect on growth at 25 °C. The effect of SA on plant height was stronger in C546 than in B125 at 4 °C. The plant height of the SA-treated seedlings increased by 19.2% in the former and by 8.7% in the latter compared with that of the non-SA seedlings. In both inbred lines, SA treatment increased the root length to the same extent (13.0% and 13.3%).

**Table 1.** Effects of SA on the growth and biomass of seedlings in two inbred lines at different temperatures.

| Inbred Lines | Temperature (°C) | SA (mg/L) | Plant Height (cm) | Root Length (cm) | Seedling DW (mg) | Root DW (mg) |
|---|---|---|---|---|---|---|
| B125 | 25 | 0 | 22.4 ± 0.5 a | 22.6 ± 1.6 ab | 130.4 ± 5.4 b | 61.5 ± 1.4 b |
| | | 50 | 23.8 ± 0.7 a | 25.0 ± 1.2 a | 152.7 ± 4.1 a | 71.0 ± 1.9 a |
| | 4 | 0 | 20.5 ± 0.6 b | 19.5 ± 0.7 c | 103.7 ± 3.6 c | 46.3 ± 1.8 c |
| | | 50 | 22.3 ± 0.4 a | 22.1 ± 1.4 b | 126.3 ± 3.9 b | 60.8 ± 4.5 b |
| C546 | 25 | 0 | 25.0 ± 0.5 ab | 24.1 ± 0.7 ab | 138.6 ± 5.2 b | 84.8 ± 2.8 b |
| | | 50 | 27.6 ± 0.8 a | 26.7 ± 1.2 a | 166.8 ± 7.9 a | 98.4 ± 6.3 a |
| | 4 | 0 | 19.7 ± 0.8 c | 20.7 ± 0.3 c | 108.5 ± 3.4 c | 61.3 ± 1.6 c |
| | | 50 | 23.5 ± 0.7 b | 23.4 ± 0.5 b | 137.4 ± 4.1 b | 84.2 ± 3.0 b |

DW—dry weight. The values show the mean ± SE. The values marked with different letters indicate significant differences between different SA concentrations within the same inbred line in each column at $p < 0.05$.

The seedling dry weight (DW) and root DW were significantly affected by the temperature, inbred line, and SA treatment ($p < 0.001$). The seedling DW and root DW decreased by 20.3% and 24.17%, respectively, in B125 and by 21.7% and 27.7%, respectively, in C546 at 4 °C compared to those at 25 °C (Table 1). At both 4 °C and 25 °C, SA significantly affected

the biomass of the inbred lines. SA increased the seedling DW and root DW more in C546 (26.6% and 37.4%) than in B125 (21.8% and 31.3%) at 4 °C.

### 3.2. Effect of SA Treatment on the Cell Membrane Permeability and MDA Content

Figure 1A shows that the REC of seedlings significantly increased at 4 °C compared to that at 25 °C. A significant difference was observed between the two inbred lines ($p < 0.001$). The REC of C546 was higher than that of B125. SA affected the REC significantly and to different extents in the two inbred lines. SA application decreased REC by 52.0% and f 23.5% in C546 and B125 at the low temperature.

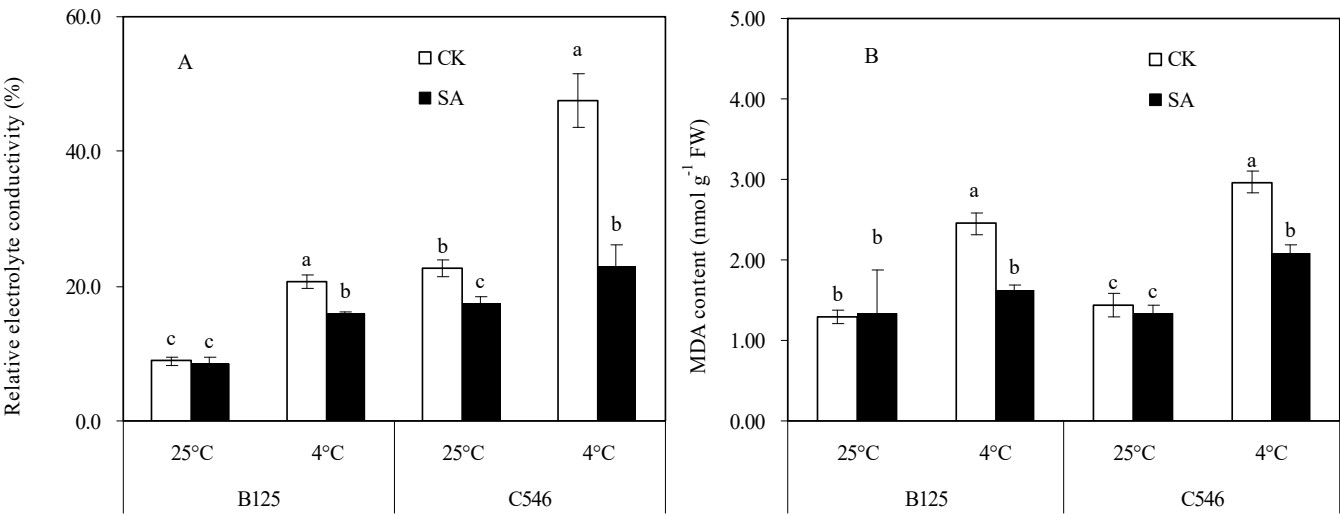

**Figure 1.** Effects of SA treatment on relative electrolyte conductivity (**A**) and malondialdehyde content (MDA) (**B**) in C546 and B125 maize seedlings. The plants were treated at 4 °C for 2 days. The different letters indicate significant differences between different SA concentrations under two temperatures within the same inbred line at $p < 0.05$.

The MDA content was significantly affected by the temperature, inbred line and SA treatment and showed significant two-factor interactions ($p < 0.001$). Figure 1B shows that the MDA content of seedlings significantly increased at 4 °C compared to 25 °C. The MDA content was significantly higher in C546 than that in B125 at the low temperature. The MDA content decreased by 29.8% and 34.0% in C546 and B125, respectively, in response to SA compared to the SA-free treatment at 4 °C. In particular, the MDA content of B125 remained the same between the low temperature and the normal temperature due to SA application.

### 3.3. Effects of SA Treatment on the Relative Water Content and Proline Content of Maize Seedlings

The RWC was affected by the temperature and the SA treatment ($p < 0.001$). RWC decreased significantly, by 54.6% in B125 and by 64.0% in C546, at 4 °C without the SA treatment (Figure 2A). There was no significant difference in the average RWC over all treatments for C546 and B125. SA significantly increased the RWC of both inbred lines at 4 °C but not that at 25 °C. The effect of SA on the RWC was greater in C546 (120.1%) than in B125 (98.3%) at 4 °C.

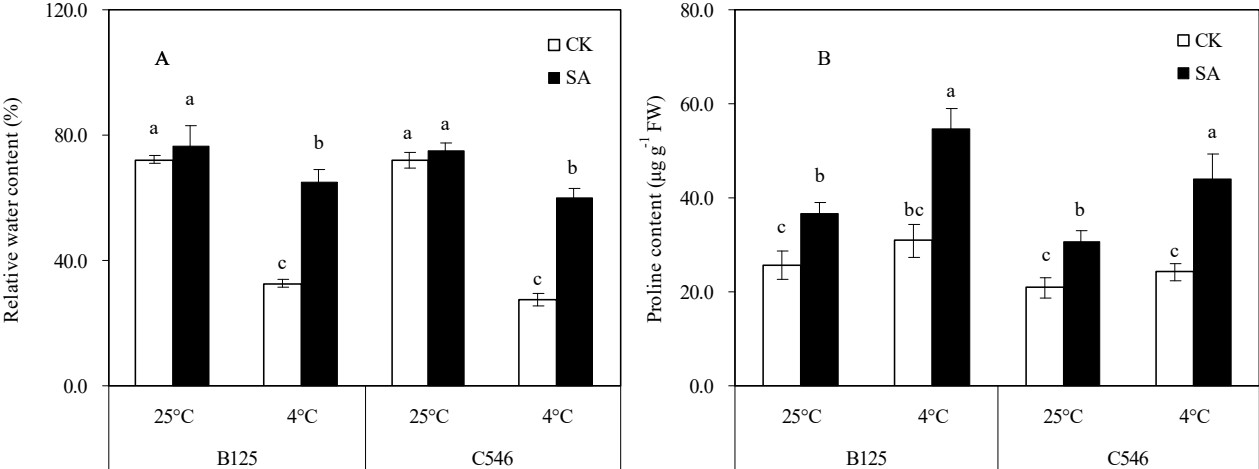

**Figure 2.** Effects of SA treatment on relative water content (**A**) and proline content (**B**) in C546 and B125 maize seedlings. The plants were treated at 4 °C for 2 days. The different letters indicate significant differences between different SA concentrations under two temperatures within the same inbred line at $p < 0.05$.

The proline content was affected by the temperature, inbred line, and SA treatment ($p < 0.001$). It significantly increased, by 20.3% in B125 and by 16.3% in C546, at 4 °C without the SA treatment (Figure 2B). The average proline content over all treatments was higher in B125 than in C546. SA significantly increased the proline content of both inbred lines at both temperatures. The effect of SA on the proline content was greater in C546 (81.4%) than in B125 (76.5%) at 4 °C.

### 3.4. Effects of SA Treatment on Maize Seedling Photosynthesis

$Pn$ decreased in both inbred lines at the low temperature, especially in C546, which was more sensitive to cold than B125 (Figure 3A). SA significantly influenced the $Pn$ at the low temperature ($p < 0.001$), and there was a significant interaction effect of SA and the inbred line ($p < 0.001$). A greater increase in $Pn$ in response to SA compared to the SA-free control was observed in C546 (40.3%) than in B125 (20.0%) at 4 °C. SA had no significant effect on $Pn$ at 25 °C.

The change trend in $Gs$ was similar to that in $Pn$. $Gs$ was significantly affected by the temperature, maize inbred line and SA treatment, and their interaction effects were significant ($p < 0.001$). $Gs$ decreased from 25 °C to 4 °C, especially in C546 (Figure 3B). At 25 °C, SA had no significant effect on $Gs$, while SA significantly influenced $Gs$ at 4 °C. The effect of SA on $Gs$ was stronger in C546 than in B125 at 4 °C.

$Ci$ was significantly affected by the temperature and SA ($p < 0.001$) but not by the maize inbred line ($p > 0.05$). $Ci$ increased from 25 °C to 4 °C (Figure 3C). SA significantly influenced $Ci$ at 4 °C but not that at 25 °C. There was no significant difference in $Ci$ between the SA treatment at 4 °C and the SA-free control at 25 °C.

The chlorophyll content was significantly affected by temperature and SA, and the effect of their interaction was significant ($p < 0.001$). The chlorophyll content decreased from 25 °C to 4 °C (Figure 3D). SA significantly increased the chlorophyll content at 4 °C but not that at 25 °C. The effect of SA on chlorophyll content was stronger in C546 (56.1%) than in B125 (50.6%) at 4 °C. There was no significant difference in chlorophyll content between the SA treatment at 4 °C and the SA-free control at 25 °C.

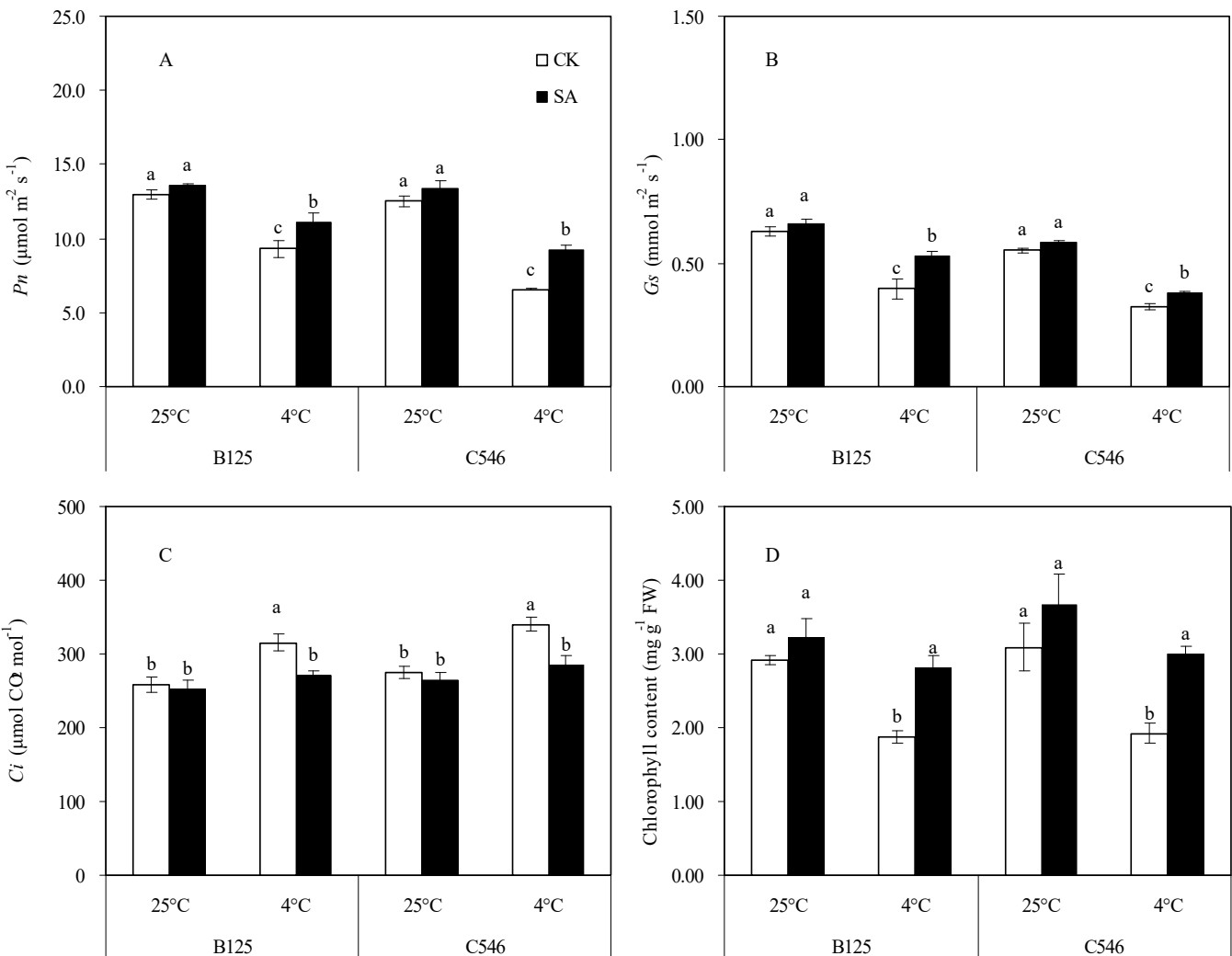

**Figure 3.** Effects of SA treatment on photosynthesis in C546 and B125 maize seedlings. (**A**) *Pn*, (**B**) *Gs*, (**C**) *Ci*, and (**D**) chlorophyll content. The plants were treated at 4 °C for 2 days. The different letters indicate significant differences between different SA concentrations under two temperatures within the same inbred line at *p* < 0.05.

### 3.5. Effects of SA Treatment on the ROS Content of Maize Seedlings

Figure 4A shows that the $H_2O_2$ content of seedlings significantly increased at 4 °C compared to 25 °C. The $H_2O_2$ content was higher in C546 than in B125 at the low temperature. The $H_2O_2$ content decreased by 16.4% in C546 and by 15.8% in B125 under the SA treatment compared to the SA-free treatment at 4 °C.

The changing trend in $O_{2-}$ production rate was similar to that for $H_2O_2$. The $O_{2-}$ production rate was significantly affected by the temperature, inbred line, and SA treatment ($p < 0.001$). Figure 4B shows that the $O_{2-}$ production rate significantly increased at 4 °C compared to 25 °C. The $O_{2-}$ production rate decreased by 21.9% in C546 and by 28.0% in B125 under the SA treatment compared to the SA-free treatment at 4 °C.

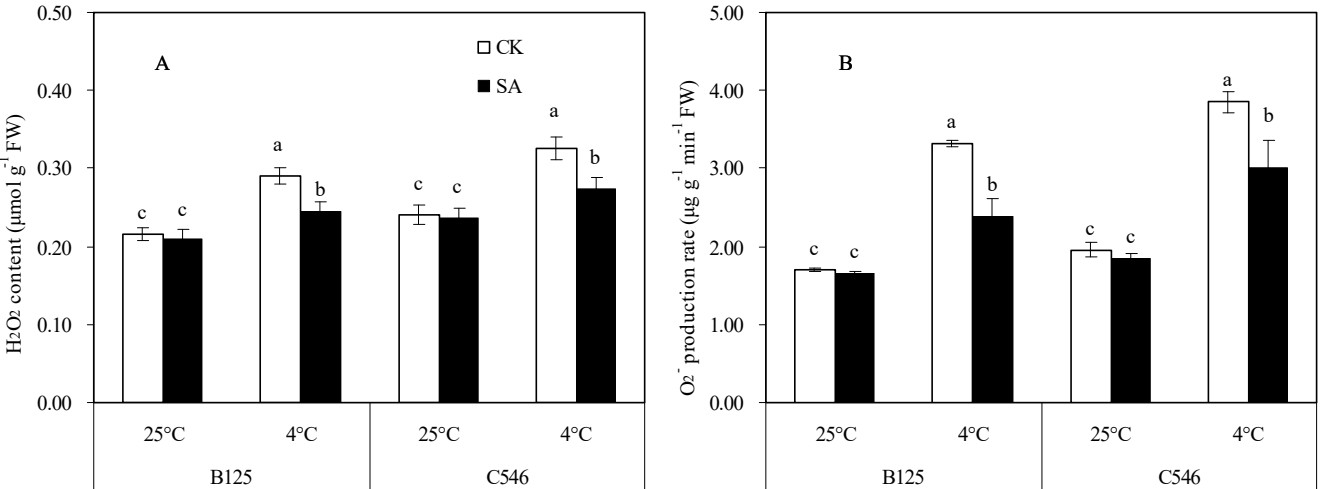

**Figure 4.** Effects of SA treatment on $H_2O_2$ content (**A**) and $O_{2-}$ production rate (**B**) in C546 and B125 maize seedlings. The plants were treated at 4 °C for 2 days. The different letters indicate significant differences between different SA concentrations under two temperatures within the same inbred line at $p < 0.05$.

### 3.6. Effects of SA Treatment on Antioxidant Enzyme System

SOD activity was significantly affected by the temperature, inbred line and SA treatment ($p < 0.001$). The effects of two-way interactions of factors were significant ($p < 0.001$). Figure 5A shows that SOD activity was significantly increased by cold stress. The increase in SOD activity in C546 (92.8%) was greater than that in B125 (67.8%) at the low temperature. The application of SA induced a significant increase in SOD activity at 4 °C compared to that in the SA-free control.

POD activity was significantly influenced by the temperature, inbred line and SA treatment ($p < 0.001$). Figure 5B shows that the POD activity significantly increased under cold stress. The increase in POD activity in C546 (160.1%) was greater than that in B125 (108.2%) at the low temperature. SA significantly increased the POD activity at 4 °C compared with that in the non-SA treatment. SA had similar effects on POD activity in both inbred lines at 4 °C.

CAT activity was significantly influenced by the temperature, inbred line and SA treatment ($p < 0.001$). Figure 5C shows that the CAT activity significantly increased under cold stress. SA significantly increased the CAT activity at 4 °C but not 25 °C.

The changing trend in APX activity was similar to that for CAT. APX activity was significantly influenced by the temperature, inbred line, and SA treatment ($p < 0.001$). Figure 5D shows that the APX activity significantly increased under cold stress. SA significantly increased the APX activity at 4 °C but not 25 °C.

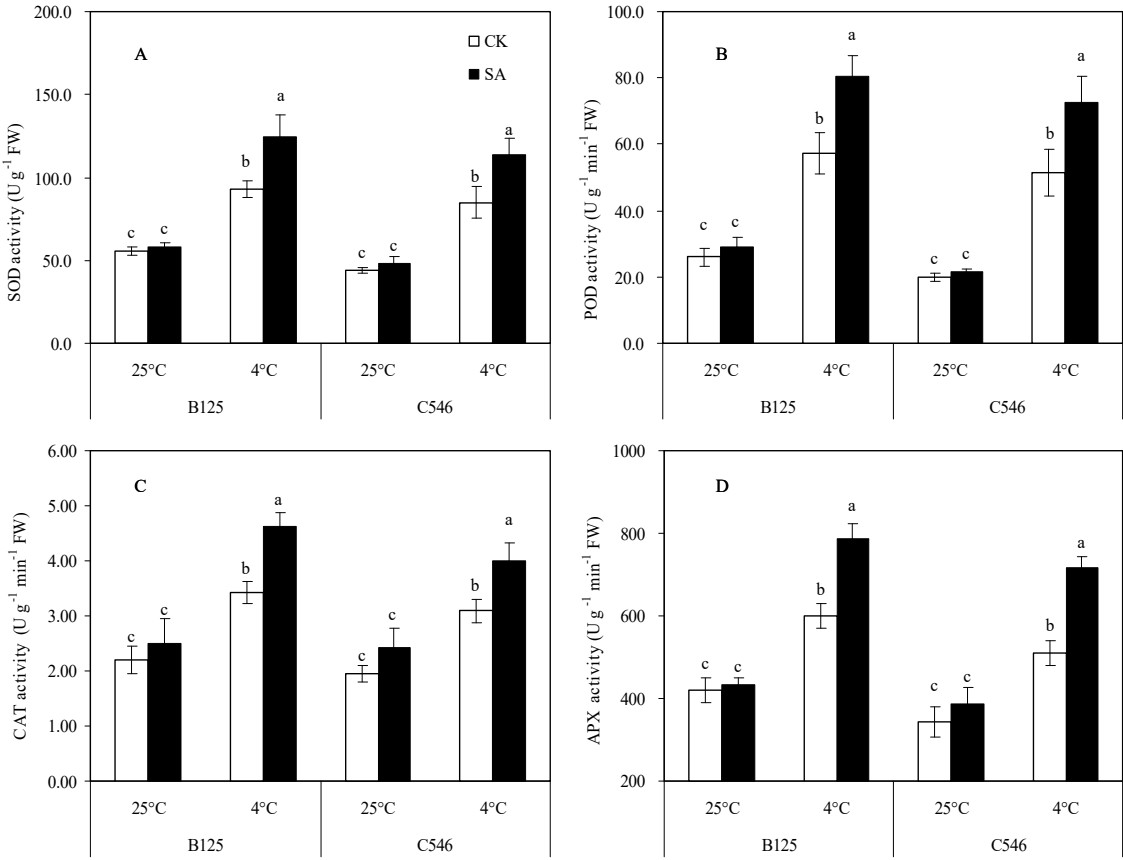

**Figure 5.** Effects of SA treatment on antioxidative enzyme activities in C546 and B125 maize seedlings. (**A**) SOD activity, (**B**) POD activity, (**C**) CAT activity, and (**D**) APX activity. The plants were treated at 4 °C for 2 days. The different letters indicate significant differences between different SA concentrations under two temperatures within the same inbred line at $p < 0.05$.

## 4. Discussion

Cold stress seriously affected the growth of different maize inbred lines in the seedling stage. Application of SA at 50 mg/L on the leaves of maize seedlings under 4 °C, promoted the seedling and root growth, resulting in enhanced chilling resistance. SA decreased the REL, the MDA content and $H_2O_2$ and $O_2^-$ content due to increased superoxide SOD, POD, CAT, and APX activity; SA also increased chlorophyll content, enhanced *Pn* and *Gs*, and decreased *Ci*. Through increasing the proline content and RWC, SA improved the osmotic adjustment capacity of seedlings. In addition, the effect of SA on the chilling resistance of C546 (cold-sensitive cv.) was stronger than that on B125 (cold-tolerant cv.) at the low temperature.

### 4.1. Growth and Physiological Changes in Maize Seedlings under Chilling Stress

Chilling stress has been recognized as an environmental factor that limits plant growth and agricultural production [33]. Short and low-biomass seedlings as well as small and yellowing leaves were observed under cold stress due to the inhibition of resource uptake processes [34]. Different organs respond differently to low temperatures. The root is an important organ with which plants absorb water and nutrients and carry out material transformation and storage. Root growth directly affects the growth and morphology of the aboveground part of the plant. Soil temperatures that are too low will inhibit root growth, resulting in short and unhealthy plants [35,36]. The shoot tip is also very sensitive to low temperatures, and this sensitivity can have a negative impact on the leaf mass and growth rate [37]. In this study, it was found that the plant height, root length and biomass of the seedlings of the two maize inbred lines were significantly affected by low temperatures;

these findings were similar to previous results [38,39]. The decreases in root growth and weight were greater than the decreases in seedling and shoot growth and weight. This suggests that the roots are more sensitive to chilling stress than other plant organs at the early growth stage.

Physiological changes occur in plants under cold stress, including primary direct and indirect damage to cell membranes. Cell membrane damage, which is the basic mechanism of low temperature stress, occurs due to the accumulation of MDA and the production of excessive ROS, such as $O_2^-$ and $H_2O_2$ which are the critical component of stress response regulation in crop plants, including maize and *Maclura pomifera* [40,41]. The primary cellular ROS generation sites are chloroplasts, mitochondria, peroxisomes, apoplast, and plasma membranes [42]. Therefore, ROS have harmful effects such as plant membrane peroxidation and intracellular electrolyte leakage, which eventually destroy the structure and function of cell membranes [43]. MDA causes the cell membrane to produce membrane lipid peroxides [44]. Li reported that the production of ROS and MDA also made the relative electrolyte conductivity (REC) increase under cold and oxidative stress. This phenomenon was also observed in our study. In this research, the MDA content and relative electrolyte conductivity (REC) in the seedlings gradually increased under chilling stress. The results also indicated that the production rate of $O_2^-$ and the content of $H_2O_2$ in the seedlings increased significantly under chilling stress. Therefore, the increase in MDA and REC may have been due to the increase in the production rate of $O_2^-$ and the content of $H_2O_2$, which would in turn lead to cell membrane integrity loss and metabolic disturbances. These findings support previous work in tomato, which showed increases in the MDA and $H_2O_2$ contents and the production rate of $O_2^-$ [45]. To reduce the accumulation of ROS and cellular oxidative damage, plants have a highly efficient antioxidant system for scavenging ROS to improve their cold tolerance [46]. In plants, the antioxidant defense system and ROS accumulation maintains a steady-state balance [43]. The antioxidant enzyme system includes SOD, POD, CAT, and APX. The results showed that the activity of the antioxidant enzymes SOD, POD, CAT, and APX in the two maize cultivars was significantly increased after the chilling stress treatment; this result is similar to the results of previous studies [38].

Maize is a typical C4 plant in which photosynthesis is sensitive to low-temperature stress [47]. Chloroplasts are the main organs for photosynthesis in plants and are one of the most cold-sensitive organs in plants. Therefore, low temperatures can damage the structure and function of chloroplasts, the net photosynthetic rate, and the gas exchange [14,39]. Bhusal confirmed that especially *Pn* and *Gs* increased due to the heavy leaf and the increase of the Chlorophyll content. This helps the leaf to capture a higher amount of light, and consequently more Chlorophyll promotes the higher of *Pn* [48]. Chen also reported that the increase of chlorophyll content also enhanced the net photosynthetic rate of maize to improve salt stress tolerance [47]. The results showed that the chlorophyll content decreased from 25 °C to 4 °C in this study. The change trends in *Pn* and *Gs* were similar to that in the chlorophyll content, while *Ci* showed a different trend at the low temperature. The same results were obtained in the study of Sun [38], who found that the *Pn, Gs*, and *Tr* of maize seedlings decreased at low temperatures, while the *Ci* increased. Aniszewski [49] and Ashraf [50] showed that the absorption of $CO_2$ and *Pn* decreases significantly or even stops when the temperature is below an optimum temperature because of plants exchanging gas through stomatal regulation to alleviate the negative effects on their photosynthetic processes. Jiang also reported that phytochromes modulated cold and freezing tolerance, and therefore the photosynthesis is closely related to low temperature [51]. *Pn* is often invoked as a reference indicator for plants responding to cold stress.

Under chilling stress, plant cells lose water, which leads to a water deficit; this results in osmotic stress, and the RWC of the cell decreases. The RWC of plant cells is an important indicator of the physiological water status in plant tissues [21]. This experiment also found that RWC significantly decreased, by 50–64%, in the two inbred lines of maize at 4 °C. This result is consistent with the research by Aroca [35]. To adapt to osmotic stress, plants adopt an active adjustment method through their own defense system to increase the

accumulation of osmotic adjustment substances, such as proline (Pro), which has been considered as a signaling substance regulating mitochondrial functions by activating ROS detoxification pathway [41]. Many studies have shown that plants can actively accumulate free proline under osmotic stress to increase the concentration of the cytosol, reduce osmotic potential, and increase their water retention capacity as stress adaptations [9,13,52,53]. This experiment also found that the content of free proline in maize leaves increased significantly under chilling stress. This is consistent with the results of research by Chen [53].

### 4.2. Effects of SA on Growth and Physiological Development

The phytohormone SA is involved in defenses against biotic and abiotic stresses as well as in plant development [54,55]. Exogenously supplied SA affects numerous aspects of plant growth and development, including seed germination, vegetative growth, root initiation and growth, fruit yield, senescence, stomatal closure, photosynthesis, and respiration [1,56]. Cold temperatures promote the accumulation of endogenous free SA and glucosyl SA in wheat and grapes [57], suggesting that SA is involved in the regulation of cold responses [56]. In this study, the application of SA was found to increase the growth and biomass of seedlings under chilling stress. The plant height, root length and dry weight of the two inbred lines were increased by the application of 50 mg/L SA. These results corroborate the findings of Szalai [9]. Farooq [4] clarified that 0.5 mM SA reduced damage to plants and promoted maize seedling growth under low-temperature conditions.

In general, low concentrations of SA may improve the antioxidant capacity of plants. Miura and Tada [56] reported that low SA concentrations stimulate the low-level accumulation of ROS as secondary signals for activating biological processes such as, e.g., enhancing the activity of cellular protective enzymes, including APX, CAT, SOD, POD, guaiacol peroxidase (GPX), and glutathione reductase (GR) [58]. Wang [59] found that the hydroponic application of SA or aspirin both alleviated the accumulation of $H_2O_2$. In our study, the $H_2O_2$ content and $O_2^-$ production rate decreased and the antioxidant enzyme (APX, CAT, SOD and POD) activity increased in two inbred lines with an SA treatment of 50 mg/L compared to those in an SA-free treatment at 4 °C. Malan [60] also reported that SA promoted the activity of antioxidant enzymes under chilling stress, thereby eliminating excess ROS. Farooq [4] found that the application of SA reduced the REL and MDA content of plant leaves as well as cell membrane peroxidation. Bandurska [61] found that SA improved the RWC in leaves by increasing the proline content. Both these results are consistent with our results, in which the SA treatment significantly increased the RWC of both inbred lines only at 4 °C and increased the proline content in both inbred lines at both temperatures.

Chilling stress can significantly reduce the *Pn* of plants, and SA can increase the *Pn* under stress conditions by increasing the chlorophyll content, affecting stomatal closure [62]. Low temperatures caused a decrease in the *Pn* of maize seedling leaves. Rivas-San Vicente and Plasencia [63] explained that the chlorophyll content of the maize seedling leaves decreased and the stomatal resistance of the leaves increased, which led to an increase in the intercellular carbon dioxide concentration. This study showed that the application of exogenous SA significantly alleviated the inhibitory effect of chilling stress on *Pn*, increased *Gs* and significantly reduced *Ci* in maize seedlings.

## 5. Conclusions

Cold stress seriously inhibits the growth and development of maize in northeast China and other high latitudes. Through this research, we confirmed the indicator related cold stress, including REC, MDA content, *Pn* and *Gs*, and $H_2O_2$ content and $O_2^-$ production rate. The application of 50 mg/L SA to the leaves of maize seedlings improves maize chilling resistance due to promoted growth, protects membrane function and structure, and improves the photosynthesis system and osmotic adjustment capacity of the two inbreds. We also find the effect of SA on the chilling resistance of cold-sensitive cv. was stronger than that on cold-tolerant cv. at the low temperature. Therefore, to study exogenous SA

and chilling stress together, by regulating different genes involved in assimilation and metabolic pathways, will be very interesting and important, based on the transcriptome results. Our study will provide novel insight into the application and breeding work of SA-induced chilling resistance in maize.

**Author Contributions:** D.Y. and Q.Z. designed the experiments and obtained funding for the research. Q.Z. and D.L. contributed to compiling and analyzing the data and wrote the manuscript. D.L., Q.W., X.Y., X.S., Y.W. and D.Q. conducted statistical analysis and performed the experimental analyses. T.X. and D.Y. participated in the data analysis and supervised the writing of the manuscript. All authors have read and agreed to the published version of the manuscript.

**Funding:** National Natural Science Foundation of China (31501251), National Key Research and Development Program of China (2018YFD0300103), Young Talents (19QC02) and Academic Elite (20XG25) of Northeast Agricultural University.

**Institutional Review Board Statement:** Not applicable.

**Informed Consent Statement:** Not applicable.

**Data Availability Statement:** Not applicable.

**Conflicts of Interest:** The authors declare no conflict of interest.

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
