# Peer review of "Exogenous Salicylic Acid Improves Chilling Tolerance in Maize Seedlings by Improving Plant Growth and Physiological Characteristics"

_agronomy, doi:10.3390/agronomy11071341_

Round 1

Reviewer 1 Report

General comments

I have read the manuscript (agro-1232592). Entitle: Exogenous Salicylic Acid Improves Chilling Tolerance in Maize Seedlings by Improving Plant Growth and Physiological Characteristics written by Qian Zhang et. al., for publication of agronomy. Author used the two inbred line of maize such as cold-sensitive cv. C546 and cold-tolerant cv. B125 and study the morphological, physiological and biochemical traits. The morphological indicators (root growth), physiologically determine the (photosynthesis, REL, and chlorophyll contents), and biochemically analysis malondialdehyde (MDA) and reactive oxygen species (ROS) (H2O2 and O2 ·−) superoxide dismutase (SOD), peroxidase (POD). Author found that the e effect of SA on the chilling resistance of C546 (cold-sensitive cv.) was stronger than that on B125 (cold-tolerant cv.) by application of salicylic acid (SA).

Research is short but interesting because of interrogation of different traits which helps to more conformation the results. This research is obvious application potential to protect the maize from the chilling injury. In this sense the manuscript is much valuable. However, I found lacking introduction and discussion sections. I found that these two section not connected well or not describe broader perspectives. Discussion section is comparatively shallow not well connect to the story. I found the lack of potential latest reference to support them in some cases in discussions. Author should present broad perspectives because abiotic stress is the big figures of this study. Author should provide the examples of other abiotic stress (salinity and drought) and present the negative effect of these stress as well in the introductory parts. One of the drawback is author only compared the single/single genotypes of maize. If author apply the exogenous SA with multiple maize-line result will be more solid and cover broad and more realistic. In discussion, author should to further discuss the result by biochemical prospective by addressing the all component of abiotic stress (metal, light, drought, chilling). Abiotic stress (e.g., chilling stress) product some chemical (ROS) that help to protein stabilization and enhance the plant to protect from the abiotic stress condition (that may be drought, flooding or chilling, salinity or heat stress). So author should connect that part as well to connect in discussion section to interpretation their results. Please see the below suggestion as well I gave some tips, please follow these all and improve your MS. Now I request this manuscript to author for MAJOR revision. Furthermore, the I request to authors for revision according to the rules of the journal and correct the bibliography.

Major suggestions

1) Introduction and discussion: author should have mention the text which is related and cover the all of abiotic stress first then slowly go on your study crop and subject matter of chilling injury. Because in this study abiotic stress is the main picture of the research. Please give the examples of abiotic stress shortly and maintained the negative effect of it in plant biology, please see below suggestions. Secondly, author did the study in two inbred line of maize among them one is cold sensitive C546 and one is cold tolerance. I recommended to the author to refer some references (if available) related to the others hormones if you find to minimize the risk of chilling injury.

2) Hypothesis of the study: Please modify and clearly presentation the hypothesis of the study. Author present that the SA applied in maize two inbred line and tested different traits to test the chilling injury. However, the hypothesis should to be very clear in the introduction sections because without appropriate literatures, questions or hypotheses in introduction section the entirely text will be unclear. Author should give the special attention and sequential presentation the background of the related components with enough literatures and connect those component with the good flow of writing. The connection of the introduction and discussion also comparatively lower. Please see in below suggestions and improved the related text accordingly.

Others comments and suggestions

3) Line no. 33

‘Before entering your subject matter of “chilling injury” Author should to mention the background of others stress as well because abiotic stress (e.g., metal and drought) because abiotic stree is the big figures of your study. For the metal stress and its related text please refer and cite this article “How plant copes with heavy metal” doi:10.1186/1999-3110-55-35 ‎2014 and for the drought stress refer “Response of drought stress in prunus sargentii and larix kaempferii ….https://doi.org/10.1016/j.foreco.2020.118099

and mention the text “drought reduced the morphological traits such as reduction of leaf size and vegetative growth, and physiological traits such as reduction of photosynthesis and stomatal conductance and alter the stem anatomical features””.

 4) Line no. 43

Are author found those literatures where application of other phenolic phytohormones which behaves as a signal for minimize the freezing injury. Or any potential previous study and related reference related to minimize the chilling injury apart from the SA. If so, please mentioned way to minimize this stress to the plants.

5) Line no. 63

“Identify which maize 63 line was more sensitive to SA application” effects of SA on the physiological 62 traits of two different maize inbred lines under cold stress it will obviously find out the more sensitive and resistance line. Therefore, please not mention the objective 1 and 2. Just simply connect those together.

6) Line no. 75

The experimental design is very simple, author just sown the seed of maize cultivar and spray the SA in control conditions and made the finding of chilling tolerance cultivars. It is not too weak experimental set up? if author set the strong design such as trial on different SA concentration or multiples line of maize cultivars or periodic application response of SA in maize something…. this hypothetical points and accordingly set the experiment design is really interesting. Unfortunately, I found very shallow findings. It is another weakness of this manuscript. These all your future insight or questions for upcoming manuscript.

7) Line no. 87

Author should to be identify the time for measurement of photosynthesis because of midday depression of photosynthesis has important consequences for ecosystem carbon exchange. It is also demonstrated that latent reduction of photosynthetic capacity can begin in the early morning, preceding the midday depression.

 8) Line no. 90-92

Please also mentioned how often you measure the photosynthesis and which instrumental set up for the instrument such as area of chamber head, and temperature in the cuvette.

9) Line no. 106

I agree with author, that if the methodology sections are request to refer to others. It is allowing in scientific writing as well, but it also depend case by case. If mythology sections are not too long it will better to present, the short procedure inside the particular sub-tile rather than this.

10) Line no. 193

Please improve the all figures slightly increase the font size in the X-and Y axis as well as the text inside the figures panel. Please check all and improve the figures visibility, because it too small.

11) Line no. 306-311

Author well mentioned the information of ROS in the discussion section in Line no. 306 to 311. However, this is still incomplete. Author should describe wide range with cover the almost all abiotic stress (include salt, drought, heat).  Please improve the text mention that “abiotic stress such as salt, drought stress plant produces the ROS when these plant exposed to the stress condition and plant produce antioxidant, flavonoids, and secondary metabolites play to role for protect the plant for detoxify ROS and protect the plant to protect the abnormal condition (i.e. stress) and protein and amino acid stabilization”. Cite the text with these reference in the paragraph somewhere (1) https://doi.org/10.1016/j.scitotenv.2021.146466 Title: Evaluation of morphological, physiological, and biochemical … (2) https://doi.org/10.1038/s41598-019-55889-Morphological, physiochemical and antioxidant responses.

12) Line no. 318

I agree with the author that “change trends in Pn and Gs were similar to that in the chlorophyll content” however author lacking to mentioned the possible reason of behind of it with the references which is very important. Please see both these articles  https://doi.org/10.1016/j.scienta.2017.12.006 and https://doi.org/10.1016/j.scienta.2013.05.036 and improved the text that “physiological performance specially Pn and Gs increased due to the leaf thickness and increase the Chlorophyll content because those help to capture the better light and higher amount of light due to Chl then higher possibility of Pn (references)”.

13) Line no. 368

Author mentioned that the Low temperatures caused a decrease in the Pn of maize seedling leaves but I not see the potential evidence in the previous study. Please mentioned that related references in maize or other crops which are chilling injured and reduced the Pn.

14) Line no. 374

Please improve the conclusion some more. Author should not the repeat the same result twice in conclusion also. Please see the line 381-383 author wrote the same text as result and introduction and result sections as well. I agree this result but please change the tone of presentation. Please note that the conclusion should to be more solid text without repetition of results and should be present the future insight of the research based on your current finding and strength of your results for the future research guideline of the related research. 

15) Line no. 393

Reference: please double check the citations, its style and spell check and other grammatical errors. moreover, I request to authors for revision throughout the manuscript according to the journal rules.

Good Luck!

Author Response

Response to Reviewer 1’s Comments

I have read the manuscript (agro-1232592). Entitle: Exogenous Salicylic Acid Improves Chilling Tolerance in Maize Seedlings by Improving Plant Growth and Physiological Characteristics written by Qian Zhang et. al., for publication of agronomy. Author used the two inbred line of maize such as cold-sensitive cv. C546 and cold-tolerant cv. B125 and study the morphological, physiological and biochemical traits. The morphological indicators (root growth), physiologically determine the (photosynthesis, REL, and chlorophyll contents), and biochemically analysis malondialdehyde (MDA) and reactive oxygen species (ROS) (H2O2 and O2 ·−) superoxide dismutase (SOD), peroxidase (POD). Author found that the e effect of SA on the chilling resistance of C546 (cold-sensitive cv.) was stronger than that on B125 (cold-tolerant cv.) by application of salicylic acid (SA).

Research is short but interesting because of interrogation of different traits which helps to more conformation the results. This research is obvious application potential to protect the maize from the chilling injury. In this sense the manuscript is much valuable. However, I found lacking introduction and discussion sections. I found that these two section not connected well or not describe broader perspectives. Discussion section is comparatively shallow not well connect to the story. I found the lack of potential latest reference to support them in some cases in discussions. Author should present broad perspectives because abiotic stress is the big figures of this study. Author should provide the examples of other abiotic stress (salinity and drought) and present the negative effect of these stress as well in the introductory parts. One of the drawback is author only compared the single/single genotypes of maize. If author apply the exogenous SA with multiple maize-line result will be more solid and cover broad and more realistic. In discussion, author should to further discuss the result by biochemical prospective by addressing the all component of abiotic stress (metal, light, drought, chilling). Abiotic stress (e.g., chilling stress) product some chemical (ROS) that help to protein stabilization and enhance the plant to protect from the abiotic stress condition (that may be drought, flooding or chilling, salinity or heat stress). So author should connect that part as well to connect in discussion section to interpretation their results. Please see the below suggestion as well I gave some tips, please follow these all and improve your MS. Now I request this manuscript to author for MAJOR revision. Furthermore, the I request to authors for revision according to the rules of the journal and correct the bibliography.

Major suggestions

1) Introduction and discussion: author should have mention the text which is related and cover the all of abiotic stress first then slowly go on your study crop and subject matter of chilling injury. Because in this study abiotic stress is the main picture of the research. Please give the examples of abiotic stress shortly and maintained the negative effect of it in plant biology, please see below suggestions. Secondly, author did the study in two inbred line of maize among them one is cold sensitive C546 and one is cold tolerance. I recommended to the author to refer some references (if available) related to the others hormones if you find to minimize the risk of chilling injury.

Thanks.

The first sentence in the second paragraph, we descripted damage caused by abiotic stress, and the plant responds including the hormones production under stress. And the several references have been cited.

Before this work, we had carried out the classification of cold resistance in inbred lines and hybred of maize, identified the optimum concentration SA from 0, 25, 50, 100 and 150 mg/L and studied the effects of different exogenous substances, such as SA, and Ca2+ .

Therefore, this research is not only seemed the simple, but also basic, practical and interesting.

2) Hypothesis of the study: Please modify and clearly presentation the hypothesis of the study. Author present that the SA applied in maize two inbred line and tested different traits to test the chilling injury. However, the hypothesis should to be very clear in the introduction sections because without appropriate literatures, questions or hypotheses in introduction section the entirely text will be unclear. Author should give the special attention and sequential presentation the background of the related components with enough literatures and connect those component with the good flow of writing. The connection of the introduction and discussion also comparatively lower. Please see in below suggestions and improved the related text accordingly.

Thanks the good suggestion.

We have rewrite hypotheses in introduction section. And revised the discussion part.

Others comments and suggestions

3) Line no. 33

‘Before entering your subject matter of “chilling injury” Author should to mention the background of others stress as well because abiotic stress (e.g., metal and drought) because abiotic stree is the big figures of your study. For the metal stress and its related text please refer and cite this article “How plant copes with heavy metal” doi:10.1186/1999-3110-55-35 ‎2014 and for the drought stress refer “Response of drought stress in prunus sargentii and larix kaempferii ….https://doi.org/10.1016/j.foreco.2020.118099

and mention the text “drought reduced the morphological traits such as reduction of leaf size and vegetative growth, and physiological traits such as reduction of photosynthesis and stomatal conductance and alter the stem anatomical features”.

Thanks very much.

however, the aim of this paper is to briefly describe the growth and physiology of maize under low temperature and SA treated. And the research parameters in our MS are not many or including the conductance and alter the stem anatomical features. We have added the content and references about abiotic stress in the discussion part and intruduction.

4) Line no. 43

Are author found those literatures where application of other phenolic phytohormones which behaves as a signal for minimize the freezing injury. Or any potential previous study and related reference related to minimize the chilling injury apart from the SA. If so, please mentioned way to minimize this stress to the plants.

Thanks for the good suggestion.

5) Line no. 63

“Identify which maize 63 line was more sensitive to SA application” effects of SA on the physiological 62 traits of two different maize inbred lines under cold stress it will obviously find out the more sensitive and resistance line. Therefore, please not mention the objective 1 and 2. Just simply connect those together.

Thanks. Have done.

6) Line no. 75

The experimental design is very simple, author just sown the seed of maize cultivar and spray the SA in control conditions and made the finding of chilling tolerance cultivars. It is not too weak experimental set up? if author set the strong design such as trial on different SA concentration or multiples line of maize cultivars or periodic application response of SA in maize something…. this hypothetical points and accordingly set the experiment design is really interesting. Unfortunately, I found very shallow findings. It is another weakness of this manuscript. These all your future insight or questions for upcoming manuscript.

We had set five concentration of SA (0, 25, 50, 100 and 150 mg/L) to select the optimal concentration before this experiment. Then the result shown that 50 mg/L was the optimal concentration.

7) Line no. 87

Author should to be identify the time for measurement of photosynthesis because of midday depression of photosynthesis has important consequences for ecosystem carbon exchange. It is also demonstrated that latent reduction of photosynthetic capacity can begin in the early morning, preceding the midday depression.

Thanks. Have done.

8) Line no. 90-92

Please also mentioned how often you measure the photosynthesis and which instrumental set up for the instrument such as area of chamber head, and temperature in the cuvette.

Thanks. Have done.

9) Line no. 106

I agree with author, that if the methodology sections are request to refer to others. It is allowing in scientific writing as well, but it also depend case by case. If mythology sections are not too long it will better to present, the short procedure inside the particular sub-tile rather than this.

Thanks. Have done.

10) Line no. 193

Please improve the all figures slightly increase the font size in the X-and Y axis as well as the text inside the figures panel. Please check all and improve the figures visibility, because it too small.

Thanks. Have done.

11) Line no. 306-311

Author well mentioned the information of ROS in the discussion section in Line no. 306 to 311. However, this is still incomplete. Author should describe wide range with cover the almost all abiotic stress (include salt, drought, heat).  Please improve the text mention that “abiotic stress such as salt, drought stress plant produces the ROS when these plant exposed to the stress condition and plant produce antioxidant, flavonoids, and secondary metabolites play to role for protect the plant for detoxify ROS and protect the plant to protect the abnormal condition (i.e. stress) and protein and amino acid stabilization”. Cite the text with these reference in the paragraph somewhere (1) https://doi.org/10.1016/j.scitotenv.2021.146466 Title: Evaluation of morphological, physiological, and biochemical … (2) https://doi.org/10.1038/s41598-019-55889-Morphological, physiochemical and antioxidant responses.

Thanks. The reviewer is so kind. We added one sentence in discussion and a reference.

12) Line no. 318

I agree with the author that “change trends in Pn and Gs were similar to that in the chlorophyll content” however author lacking to mentioned the possible reason of behind of it with the references which is very important. Please see both these articles  https://doi.org/10.1016/j.scienta.2017.12.006 and https://doi.org/10.1016/j.scienta.2013.05.036 and improved the text that “physiological performance specially Pn and Gs increased due to the leaf thickness and increase the Chlorophyll content because those help to capture the better light and higher amount of light due to Chl then higher possibility of Pn (references)”.

 Thanks. We added one sentence in discussion and a reference to confirm our result.

13) Line no. 368

Author mentioned that the Low temperatures caused a decrease in the Pn of maize seedling leaves but I not see the potential evidence in the previous study. Please mentioned that related references in maize or other crops which are chilling injured and reduced the Pn.

Thanks. The references of Su (2020) and Aniszewski (2001) found when the temperature is below an optimum temperature, the absorption of CO2 and Pn declined significantly, or even stops. Therefore, Pn can be invoked as a reference indicator for plant.

14) Line no. 374

Please improve the conclusion some more. Author should not the repeat the same result twice in conclusion also. Please see the line 381-383 author wrote the same text as result and introduction and result sections as well. I agree this result but please change the tone of presentation. Please note that the conclusion should to be more solid text without repetition of results and should be present the future insight of the research based on your current finding and strength of your results for the future research guideline of the related research. 

Thanks. Have done.

15) Line no. 393

Reference: please double check the citations, its style and spell check and other grammatical errors. moreover, I request to authors for revision throughout the manuscript according to the journal rules.

Thanks. Have done.

Reviewer 2 Report

Comments on agronomy-1232592

Abstract

“Exogenous SA” may be replaced with “foliar applied SA”

The results should be presented quantitatively with the percentage increase or decrease in different parameters

Introduction

A clear hypothesis is missing in the last section of the introduction

Why the present study is important or what this adds to the previous knowledge is missing especially regarding physiological understandings

Materials and Methods

The abbreviation of liter in most of the places (ml, mol/l) is not standard, it should be replaced with “L”

Duncan’s test should be replaced with Duncan’s Multiple Range test

Results

The lettering on Figure 1A (25 °C C546) and 1B (25 °C C546) looks incorrect, please revisit   

Similarly with 3A, 3B

Overall, the introduction and Discussion should be improved with the latest papers from 2021

Author Response

Response to Reviewer 2’s Comments

  1. Abstract: “Exogenous SA” may be replaced with “foliar applied SA”; The results should be presented quantitatively with the percentage increase or decrease in different parameters

Thanks. Have done.

  1. Introduction: A clear hypothesis is missing in the last section of the introduction; Why the present study is important or what this adds to the previous knowledge is missing especially regarding physiological understandings

Thanks for this good suggestion. Have done.

  1. Materials and Methods: The abbreviation of liter in most of the places (ml, mol/l) is not standard, it should be replaced with “L”; Duncan’s test should be replaced with Duncan’s Multiple Range test

Have done.

  1. Results: The lettering on Figure 1A (25 °C C546) and 1B (25 °C C546) looks incorrect, please revisit; Similarly with 3A, 3B

Thanks. The information of the different letters about significant difference did not explain. 

Have done now.

  1. Overall, the introduction and Discussion should be improved with the latest papers from 2021

Thanks.

We have revised the introduction, discussion and conclusion section.

Round 2

Reviewer 1 Report

General comments

I have read the revised manuscript (agro-1232592R1). Entitle: Exogenous Salicylic Acid Improves Chilling Tolerance in Maize Seedlings by Improving Plant Growth and Physiological Characteristics written by Qian Zhang et. al., for publication of agronomy. Author used the two inbred line of maize such as cold-sensitive cv. C546 and cold-tolerant cv. B125 and study the morphological, physiological and biochemical traits.

Author address some of the comment well but author skip some of the suggestions and their revision is is still quite not enough. The main drawback of this manuscript have lack of nobility. Author already tested the different concentration of SA in their previous paper and in this paper just single concentration of SA tested the morphology and physiology which is not more cover the new thing and that cause this paper have lacking the nobility. Please, revision means not only revised the text those indicated by the reviewers but revision should to be based on the gentle modified the manuscript thoroughly based on the reviewer comments. This skill should to develop by researchers while writing the scientific paper. However, I not found this in this revision. Author directly give the answer of the without consider the manuscript flow and arranging the all necessary components. Moreover, I request to author to follow and refer some of the articles author careless to this regards. Manuscript is improving than the before but there is also many space to further corrections please. However, I found lacking introduction and discussion sections still happen. I found that these two section not connected well or not describe broader perspectives. Discussion section is comparatively shallow not well connect to the story. In discussion, I was suggested to author should to further discuss the result by biochemical prospective by addressing the all component of abiotic stress (metal, light, drought, chilling). Abiotic stress (e.g., chilling stress) product some chemical (ROS) that help to protein stabilization and enhance the plant to protect from the abiotic stress condition (that may be drought, flooding or chilling, salinity or heat stress). However, still this is not satisfactory. Now I request this manuscript to author for MAJOR revision.

Please address these comments

1) Line no. 33 (original draft)

‘Before entering your subject matter of “chilling injury” Author should to mention the background of others stress as well because abiotic stress (e.g., metal and drought) For the metal stress and its related text please cite this article “How plant copes with heavy metal” doi:10.1186/1999-3110-55-35 ‎2014 and for the drought stress cite this article “Response of drought stress in prunus sargentii and larix kaempferii ….https://doi.org/10.1016/j.foreco.2020.118099

and mention the text “drought reduced the morphological traits such as reduction of leaf size and vegetative growth, and physiological traits such as reduction of photosynthesis and stomatal conductance and alter the stem anatomical features””.

2) Line no. 75

The experimental design is still very simples, author already tested the different concentration of SA in their previous study so please give the novility for this study. Otherwise it seems like very simple, author just sown the seed of maize cultivar and spray the SA in control conditions and made the finding of chilling tolerance cultivars. This is very simple and shallow finding.

3) Line no. 106

I agree with author, that if the methodology sections are request to refer to others. It is allowing in scientific writing as well, but it also depend case by case. If methodology sections are not too long it will better to present, the short procedure inside the particular sub-tile rather than this.

4) Line no. 306-311

Author well mentioned the information of ROS in the discussion section in Line no. 306 to 311 in original manuscript. However, author did not cite these literature in their manuscript revisions and their related text please see the original comment (1) https://doi.org/10.1016/j.scitotenv.2021.146466 Title: Evaluation of morphological, physiological, and biochemical … (2) https://doi.org/10.1038/s41598-019-55889-Morphological, physiochemical and antioxidant responses.

4) Line no. 318 original manuscipt (Please also address the previous comment no. 12)

I agree with the author that “change trends in Pn and Gs were similar to that in the chlorophyll content” however author lacking to mentioned the possible reason of behind of it with the references which is very important. Please see both these articles  https://doi.org/10.1016/j.scienta.2017.12.006 and https://doi.org/10.1016/j.scienta.2013.05.036 and improved the text that “physiological performance specially Pn and Gs increased due to the leaf thickness and increase the Chlorophyll content because those help to capture the better light and higher amount of light due to Chl then higher possibility of Pn (references)”.

5) Line no. 368

Author mentioned that the Low temperatures caused a decrease in the Pn of maize seedling leaves but I not see the potential evidence in the previous study. Please mentioned that related references in maize or other crops which are chilling injured and reduced the Pn.

6) Line no. 374

Please improve the conclusion still some more. Here is still repetation of the same result in above. Please present the future insight of the research based on your current finding and strength of your results for the future research guideline of the related research. 

Author Response

Please address these comments

1) Line no. 33 (original draft)

‘Before entering your subject matter of “chilling injury” Author should to mention the background of others stress as well because abiotic stress (e.g., metal and drought) For the metal stress and its related text please cite this article “How plant copes with heavy metal” doi:10.1186/1999-3110-55-35 ‎2014 and for the drought stress cite this article “Response of drought stress in prunus sargentii and larix kaempferii ….  https://doi.org/10.1016/j.foreco.2020.118099 and mention the text “drought reduced the morphological traits such as reduction of leaf size and vegetative growth, and physiological traits such as reduction of photosynthesis and stomatal conductance and alter the stem anatomical features””.

We have revised the MS according the reviewer comments and cited the two references.

We added the first paragraph in the introduction where the background of others stress as well as temperature, drought, waterlogging, salinity and heavy metals have mentioned.

2) Line no. 75

The experimental design is still very simples, author already tested the different concentration of SA in their previous study so please give the novility for this study. Otherwise it seems like very simple, author just sown the seed of maize cultivar and spray the SA in control conditions and made the finding of chilling tolerance cultivars. This is very simple and shallow finding.

I agree with the reviewer that the design is simples. But this research is continuous and very important. We think it is important that the clearly results and conclusion through the simple design.

3) Line no. 106

I agree with author, that if the methodology sections are request to refer to others. It is allowing in scientific writing as well, but it also depend case by case. If methodology sections are not too long it will better to present, the short procedure inside the particular sub-tile rather than this.

Firstly, I did not understand the reviewer meaning in the previous version. I thought we need briefly and scientific writing in the method part.

This time I feel that: the reviewer think that there are too many sub-title and too many paragraphs. If I feel right, I have delete the third-level sub-tile.

4) Line no. 306-311

Author well mentioned the information of ROS in the discussion section in Line no. 306 to 311 in original manuscript. However, author did not cite these literature in their manuscript revisions and their related text please see the original comment (1) https://doi.org/10.1016/j.scitotenv.2021.146466 Title: Evaluation of morphological, physiological, and biochemical … (2) https://doi.org/10.1038/s41598-019-55889-Morphological, physiochemical and antioxidant responses.

I am sorry that we did not cite the two references which are very good researches and well writing, in the previous comment. There were two reasons. The firstly, the plants involved in these two references are trees not crops whose physiological process and response to abiotic stress are very different. The secondly, the two references are all about drought not cold stress which is the key and theme.

We have cited this reference “Morphological, physiochemical and antioxidant responses……. …. ” which is better.

4) Line no. 318 original manuscipt (Please also address the previous comment no. 12)

I agree with the author that “change trends in Pn and Gs were similar to that in the chlorophyll content” however author lacking to mentioned the possible reason of behind of it with the references which is very important. Please see both these articles  https://doi.org/10.1016/j.scienta.2017.12.006 and https://doi.org/10.1016/j.scienta.2013.05.036 and improved the text that “physiological performance specially Pn and Gs increased due to the leaf thickness and increase the Chlorophyll content because those help to capture the better light and higher amount of light due to Chl then higher possibility of Pn (references)”.

We receive the reviewer’s suggestion. In our research, the Pn and Gs increased with the higher Chl content and dry weight (DW) of the seedlings.

We have cited one reference recommended by reviewer.

5) Line no. 368

Author mentioned that the Low temperatures caused a decrease in the Pn of maize seedling leaves but I not see the potential evidence in the previous study. Please mentioned that related references in maize or other crops which are chilling injured and reduced the Pn.

The references of Su (2020) and Aniszewski (2001) found when the temperature is below an optimum temperature, the absorption of CO2 and Pn declined significantly, or even stops. The potential evidence is that “the primary cellular ROS generation sites are chloroplasts, mitochondria, etc., therefore ROS have harmful effects to structure and function of chloroplasts under cold stress”. Chloroplast is damaged, the chlorophyll content is reduced so that Pn decline. Therefore, Pn can be invoked as a reference indicator for plant.

In addition, Phytochromes, which are essential for the full development of cold acclimation in Arabidopsis thaliana, were shown to modulate cold and freezing tolerance and help plants better adapt to harsh environmental conditions. (The reference title: Cold-Induced CBF–PIF3 Interaction Enhances Freezing Tolerance by Stabilizing the phyB Thermosensor in Arabidopsis.    https://doi.org/10.1016/j.molp.2020.04.006)

6) Line no. 374

Please improve the conclusion still some more. Here is still repetation of the same result in above. Please present the future insight of the research based on your current finding and strength of your results for the future research guideline of the related research. 

OK. Please look at the rewrite conclusion.

However, we think it is necessary to re-mention the research results, thereby we put the content in the first paragraph of discussion.

Reviewer 2 Report

The manuscript has been improved much and can be considered for publication 

Author Response

None